# Attributes and factors associated with long covid in patients hospitalized for acute COVID-19: A retrospective cohort study

Bethlehem Berhanu Minassie[1]*, Wondwossen Amogne Degu[1], Eyob Kebede Etissa[3‡], Natnael Fitsum Asfeha[2‡], Salem Taye Alemayehu[4‡], Dawit Kebede Huluka[1]

1 Department of Internal Medicine, College of Health Sciences, Addis Ababa University, Addis Ababa, Ethiopia, 2 Department of Internal Medicine, Hallelujah General Hospital, Addis Ababa, Ethiopia, 3 Department of Public Health, Addis Ababa University, Addis Ababa, Ethiopia, 4 Department of Public Health, St. Paul's Hospital Millennium Medical College, Addis Ababa, Ethiopia

☯ These authors contributed equally to this work.
‡ EKE, NFA and STA also contributed equally to this work.
* bethleheminassie@gmail.com

**Data Availability Statement:** All data are available from the figshare database (https://doi.org/10.6084/m9.figshare.26605462.v1).

## Abstract

### Background

It is now recognized that many patients have persistent symptoms after recovery from acute COVID-19 infection, an infection caused by the coronavirus SARS-CoV-2. This constellation of symptoms known as 'Long COVID' may manifest with a wide range of physical and cognitive/psychological symptoms. Few data are available on the prevalence, attributes, and factors associated with Long COVID in Africa.

### Method

This was a retrospective review of patients' electronic medical records from Hallelujah General Hospital (one of the first private hospitals to treat COVID-19 patients). The hospital's database was searched for patients hospitalized for acute COVID-19 infection from March 2020 to December 2022. Two hundred and forty-seven participants who underwent follow-up beginning four weeks after symptom onset were assessed for Long COVID. Admission and follow-up data were collected using Kobo Toolbox and exported into SPSS 27 for analysis. The relationship between the independent and dependent variables was explored through binary logistic regression.

### Results

One hundred seventy-eight (72.1%) participants had at least one persisting symptom 4 weeks post-symptom onset, at a median follow-up time of 35 (IQR 32–40) days. The most frequently reported symptoms were fatigue (41.7%), shortness of breath (31.2%), cough (27.1%), and sleep disturbances (15%). Duration of symptoms more than 7 days before admission [aOR = 1.97; CI$_{95\%}$ = 1.04 to 3.75; P = 0.038] and length of stay more than 10

**Funding:** Funding was obtained from College of Health Sciences, Addis Ababa University. The funder did not play any role in the study design, data collection and analysis, decision to publish, or preparation of the manuscript.

**Competing interests:** The authors have declared that no competing interests exist.

days in the hospital [aOR = 2.62; $CI_{95\%}$ = 1.20 to 5.72; P = 0.016] were found to be significantly associated with Long COVID.

## Conclusion

There is a high prevalence of Long COVID among patients hospitalized for acute COVID-19. Those who had a longer duration of symptoms before admission and a longer stay in the hospital appear to have a higher risk.

## Introduction

Since the global COVID-19 pandemic started in 2020, caused by the coronavirus SARS-CoV-2, there have been more than 775 million confirmed cases globally and more than 7 million deaths [1]. This resulted in many people recovering from this disease, but the full consequences of the infection after recovery from acute illness were not initially recognized well by the scientific community. After the term "Long COVID" was first used by a patient early in the pandemic, recognition of the condition grew over a few months, mainly through patient-led efforts [2]. Since then, various other terms have been developed to describe a range of residual symptoms experienced by patients, such as 'post-acute sequelae of COVID-19', 'post-acute sequelae of SARS-CoV-2 infection', 'chronic COVID syndrome', 'long-haul COVID', and 'post-COVID-19' [3].

Various groups have tried to define and categorize the range of symptoms experienced by COVID-19 patients during the acute and prolonged phases of the illness. The definitions share some of the following features for Post COVID conditions proposed by the Centers for Disease Control and Prevention (CDC) and World Health Organization (WHO); (1) signs and symptoms that develop in an individual with confirmed SARS-Cov-2 infection, (2) these continue four weeks or more after the initial infection, (3) multisystemic symptoms, (4) symptoms may be relapsing and remitting or progressing over time, (5) not explained by an alternative diagnosis [4, 5].

A meta-analysis published in January 2023 that included 194 studies and 735,006 participants reported that at a mean follow-up duration of 126 days, 45% of the patients continued to experience at least one ongoing symptom [3]. Another large meta-analysis of 54 studies and two medical record databases of 1.2 million individuals found that 6.2% of patients who had symptomatic COVID-19 had at least 1 of 3 symptom clusters 3 months after the infection (cognitive problems, fatigue with bodily pain or mood swings, respiratory symptoms) [6]. This vast difference in prevalence is also reflected in the numbers presented in other studies, with prevalence estimates of 6.2% - 82.1% [3, 6–14].

Fatigue is a dominant symptom of Long COVID, with prevalence estimates ranging from 3.7% to 58% in various studies [3, 6, 8–14]. Other common symptoms include respiratory problems such as shortness of breath, abnormal lung imaging or diffusing capacity of the lungs for carbon monoxide (DLCO) findings, concentration and memory problems, and disturbed sleep. [3, 6, 10–12]. Decreased quality of life is also reported in many patients [10]. In addition, it is worth noting that symptoms can be present in any system, such as hematologic, cardiovascular, renal, endocrine, gastrointestinal, hepatobiliary, and dermatologic [14, 15].

Abnormal chest X-ray/computed tomography [CT] is another persistent finding after recovery from COVID-19 (34% prevalence in one meta-analysis of 15 studies) [8]. Some common features include ground-glass opacity, evidence of fibrosis, consolidation, and reticulation

[9, 10]. Although these changes may persist for up to 90 days after discharge in two-thirds of the patients [8], improvement or radiologic resolution is observed compared to initial CT findings in some studies [10].

Potential risk factors for Long COVID have been proposed in the literature since not all patients with acute COVID-19 infection develop persistent symptoms. Some of these include female sex [6, 8–10, 13, 14], older age [9], hospital admission especially to the Intensive Care Unit (ICU) [3, 9, 10, 13], comorbidities such as asthma, hypertension, chronic lung conditions [9, 14], need for oxygen during the acute phase of the illness [9]. On the other hand, other studies found no association with sex and age [3]. Average follow-up time was also found to determine the prevalence of symptoms, with fewer patients having symptoms at 12 months compared to 3 months [6, 13].

Only a few studies have been done in Africa on Long COVID, namely in Egypt, Ghana, and South Africa [7, 16, 17]. The findings in two of these studies are consistent with those from other parts of the world: females are more affected, fatigue and respiratory symptoms are the commonest sequelae, and comorbidities such as hypertension and diabetes mellitus determine Long COVID [7, 16]. Contrary to other studies, the study done in Egypt reported more Long COVID symptoms in non-hospitalized patients than hospitalized patients [17].

In Ethiopia, only one study has been done that assessed post-COVID-19 pulmonary complications among recovered COVID-19 patients [18]. In that study, 14.1% of the patients were found to have pulmonary complications at 3 months post-hospital discharge. No other published studies are reporting on Long COVID in other systems.

Overall, insufficient data exists from Africa on Long COVID, and especially from Ethiopia. It is essential to identify patients affected by this condition better, understand how often patients develop it, and pinpoint its risk factors so that the consequences it leads to can be adequately addressed. There is a global interest in gaining an increased understanding of Long COVID, and this research adds valuable input to the existing research by filling the knowledge gap in underserved regions like Africa.

## Methods

### Study design and setting

The study was done at Hallelujah General Hospital, Addis Ababa, Ethiopia. It is one of the first private hospitals where patients with COVID-19 were treated. It has one full-time pulmonologist, four internists, one infectious diseases specialist, two part-time pulmonary and critical care physicians, and other health care professionals. It has a 6–7 ICU bed capacity. Its outpatient follow-up clinics are staffed by pulmonary and critical care as well as infectious diseases specialists. The data was taken from medical records of patients admitted for acute COVID-19 between March 2020 and December 2022 [data accessed November 1, 2023 –November 30, 2023].

### Sampling procedure and eligibility criteria

The sample size was calculated to be 226 by using the formula for single population proportion considering a prevalence of 82.1%, from a study done in South Africa in a similar population [7] and assuming a confidence interval of 95% and 5% margin of error. This was the minimum number of patients planned to be included for the results to be statistically significant. To increase the power of the study, all patients who met the eligibility criteria—namely, age above 18, having a confirmed symptomatic SARS-CoV-2 infection, and having at least one follow-up visit four weeks after the onset of symptoms—were included. Patients whose follow-up ended before four weeks have elapsed from symptom onset were excluded from the study.

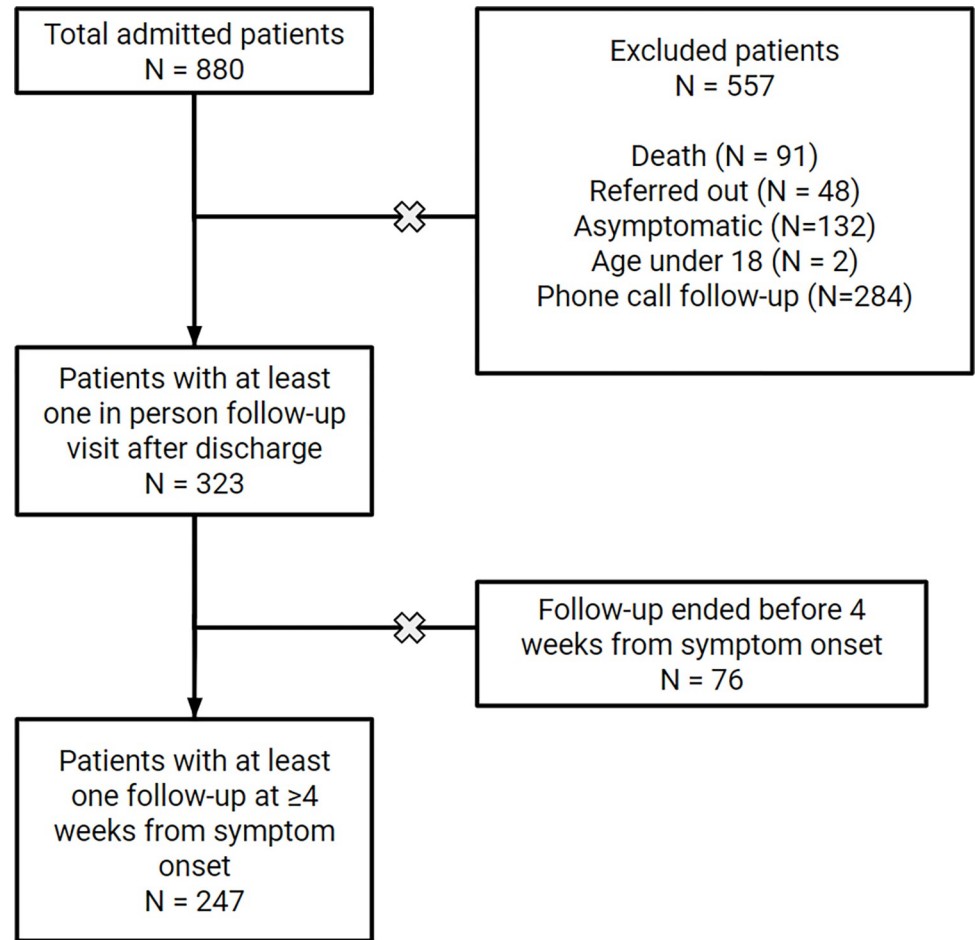

**Fig 1. Flow diagram of patient selection.**

Between March 2020 and December 2022, a total of 880 patients were admitted to Hallelujah General Hospital for acute COVID-19 treatment. Out of these, 739 patients were discharged from the hospital after completing their treatment/isolation period, and 247 of the discharged patients had in-person follow-up visit after four weeks. Among the remaining patients, 132 never developed any symptoms of COVID-19 and were not included in the study. And an additional 284 patients had post-discharge follow-up phone call and were reported to be asymptomatic, and hence were not appointed for further in-person visits (Fig 1).

## Data collection and quality assurance

Data was collected from Electronic Medical Records (EMR) for retrospective review. A structured and pretested checklist was developed to collect the data, based on WHO and CDC guidelines, established outcome sets of Long COVID [19], and information from previous research on the condition. It had 3 sections outlining the patients' socio-demographic characteristics, acute illness (clinical, laboratory, and imaging features, complications, treatments received), and data recorded during their follow-up (S2 File). The data was collected by trained general practitioners using Kobo toolbox.

## Statistical analysis

After data collection, it was thoroughly cross-checked and exported to the SPSS software version 27 (New York, USA) for analysis. Continuous data were described using the median and interquartile range (IQR) after skewness was checked, while categorical variables were provided as frequency and percentage (%). Factors associated with Long COVID were determined by unadjusted logistic regression and those with a P value of <0.25 were considered for multivariable analysis. Statistical significance was considered at P < 0.05. Model fitness was checked using the Hosmer-Lemeshow goodness of fit test.

## Operational definitions

Long COVID was defined in this study as signs and symptoms that develop in an individual with confirmed SARS-Cov-2 infection that continue four weeks or more after the initial infection. Time zero was taken as the date of symptom onset of acute COVID-19 infection. Chest CT findings were classified based on Radiological Society of North America categorization (S1-4 Table in S1 File) [20]. The classification of acute COVID-19 infection severity was based on the national COVID-19 management guideline (S1-5 Table in S1 File) [21].

## Ethical considerations

Ethical approval was obtained from the Addis Ababa University, College of Health Sciences Institutional Review Board (IRB) [Letter reference number 45/24, protocol number 44/22]. Data was anonymized, and the confidentiality of the participants was maintained throughout the study. The authors had no access to information that could identify individual participants during the study.

## Results

### Socio-demographic characteristics

The mean age of the participants was 58.5 (±13.9) years. One-third of the participants were above the age of 65 (Table 1). Most of the patients were male with a male-to-female ratio of 1.8:1. One hundred and one (40.9%) participants were either overweight or obese.

**Table 1. Sociodemographic characteristics of the participants.**

| Age | Frequency (n) | Percentage (%) |
|---|---|---|
| 18–34 | 7 | 2.8 |
| 35–49 | 65 | 26.3 |
| 50–64 | 89 | 36 |
| ≥65 | 86 | 34.8 |
| **Sex** | | |
| Male | 160 | 64.8 |
| Female | 87 | 35.2 |
| **BMI** | | |
| Normal | 28 | 11.3 |
| Overweight | 45 | 18.2 |
| Obese | 56 | 22.7 |
| Not stated | 118 | 47.8 |
| **Total** | 247 | 100 |

BMI: Body mass index

## Characteristics of the patients during admission for acute COVID-19 infection

Only one patient had a re-infection, whereas it was a first-time infection for 246 (99.6%) patients. Ten patients (4%) had received COVID-19 vaccination before their infection. The median duration of viral shedding (time from the first positive COVID-19 PCR to the first negative test) was 14 days (IQR 11–17). Patients had five symptoms on average during their acute illness, and they experienced their symptoms for a median of 7 days before admission (Table 2).

Patients were admitted for a median of 13 days (IQR 10–15). One hundred seventy-seven (71.7%) patients were admitted to the hospital for more than 10 days. Twenty-two (8.9%) patients were admitted to the ICU during their stay. While 48 (19.4%) did not need any oxygen support, 13 (5.6%) patients required a higher level of respiratory support i.e. noninvasive ventilation or mechanical ventilation (Table 3). Eighteen (7.3%) patients received antiviral treatment (Remdesivir).

One hundred and forty-eight (59.9%) patients had at least one comorbidity. The most common of these include diabetes mellitus (34%), hypertension (33.6%), and dyslipidemia (10.1%) (S1-1 Table in S1 File). Twenty-two (8.9%) patients had 2 or more complications during admission, while 123 (49.8%) patients had no complications at all (S1-2 Table in S1 File). The most common complication was secondary bacterial infection, experienced by 97 (39.3%) patients.

## Follow-up characteristics

Among the patients who had follow-up at least 4 weeks after the onset of symptoms, the median timing of the first follow-up was 35 (IQR 32–40) days, and that of the last follow-up was 43 (IQR 34–65) days. Data on clinical findings and laboratory and imaging abnormalities recorded during the follow-up was collected, to look for any persisting abnormalities. One hundred seventy-eight (72.1%) patients had at least one persisting symptom 4 weeks post-symptom onset. Patients had 2 symptoms on average, the most frequently reported symptoms being fatigue (41.7%), shortness of breath (31.2%), cough (27.1%), and sleep disturbances (15%) (Table 4). During their last follow-up (at a median of 43 days), 62.5% of the patients had persistent symptoms.

Among the ten patients who were vaccinated, six had persistent symptoms during their follow-up. Nine patients (3.6%) had $SpO_2$ less than 90% on room air, out of which 2 patients had a preexisting cardiopulmonary illness.

Lymphopenia was present in 44 (17.8%) patients, while 36 (14.6%) patients had leukocytosis (S1-3 in Table S1 File,). During their follow-up, 68 (27.5%) patients had chest X-rays done, out of whom 12 (17.6%) had normal readings. Among patients for whom chest X-ray was obtained, the most common abnormality was fibrosis, detected in 29 (42.6%) patients, followed by ground-glass opacities in 27 (39.7%) patients. Additional abnormalities included pleural effusion in one patient, consolidation in two patients, and reticulations in six patients. Only one patient had a chest CT done, and it showed residual fibrotic bands with acute pulmonary thromboembolism.

## Factors associated with Long COVID in patients hospitalized for acute COVID-19 infection

While patients with age $\geq$70 were found to have 2.2 times increased odds of having Long COVID on bivariate analysis [$CI_{95\%}$ = 1.06, 4.72], this was not true when controlling for the

**Table 2. Characteristics of the patients during admission.**

| | | Frequency (n) | Percentage (%) |
|---|---|---|---|
| **Severity of acute COVID-19 infection** | Mild | 5 | 2 |
| | Moderate | 27 | 10.9 |
| | Severe | 196 | 79.4 |
| | Critical | 19 | 7.7 |
| **Number of symptoms** | <5 | 116 | 47 |
| | ≥5 | 131 | 53 |
| **Duration of symptoms before hospitalization** | ≤7 days | 130 | 52.6 |
| | >7 days | 117 | 47.4 |
| **Symptoms** | Cough | 201 | 81.4 |
| | Fatigue | 166 | 67.2 |
| | Fever | 133 | 53.8 |
| | Dyspnea | 115 | 46.6 |
| | Myalgia/arthralgia | 85 | 34.4 |
| | Chills | 47 | 19 |
| | Headache | 58 | 23.5 |
| | Chest pain | 25 | 10 |
| | Diarrhea | 23 | 9.3 |
| | Nausea/vomiting | 23 | 9.3 |
| | Anosmia | 47 | 19 |
| | Ageusia | 51 | 20.6 |
| | Loss of appetite | 84 | 34.1 |
| **Laboratory abnormalities** | Anemia (Hgb < 13 g/dL (M), <12 g/dL (F)) | 50 | 20.2 |
| | Leukocytosis (WBC > 11,000/μL) | 152 | 61.5 |
| | Lymphopenia (Lymphocyte count <1000/μL) | 212 | 85.8 |
| | Raised aminotransferases (AST >38 U/L, ALT >63 U/L) | 82 | 33.2 |
| | Raised LDH (>450 U/L) | 55 | 22.3 |
| | Raised ESR (>15 mm/hr) | 58 | 23.5 |
| | Raised CRP (>20 mg/L) | 94 | 38.1 |
| | Raised creatinine (>1.3 mg/dL (M), >0.9 mg/dL (F)) | 34 | 13.8 |

| **Imaging abnormalities** | Chest x-ray | | Chest CT | |
|---|---|---|---|---|
| | Frequency (n) | Percentage (%) | Frequency (n) | Percentage (%) |
| Typical | 212 | 85.8 | 76 | 30.8 |
| Atypical | 22 | 8.9 | 6 | 2.4 |
| Indeterminate | 4 | 1.6 | - | |
| No features of pneumonia | 6 | 2.4 | 1 | 0.4 |
| Not done | 3 | 1.2 | 164 | 66.4 |

Abbreviations: ALT—Alaline aminotransferase, AST—Aspartate aminotransferase, CRP–C-reactive protein, ESR–Erythrocyte sedimentation rate, F–Female, Hgb–Hemoglobin, LDH–Lactate dehydrogenase, M–Male, WBC–White blood cells

other variables (Table 5). Other factors such as viral shedding more than 2 weeks, duration of symptoms >7 days before admission, length of stay >10 days in the hospital, and staying an extra day in the hospital were found to confer risk of Long COVID on bivariate analysis. However, only duration of symptoms >7 days before admission [adjusted odds ratio [aOR] = 1.97;

**Table 3. Treatment received by the patients.**

| | | Frequency (n) | Percentage (%) |
|---|---|---|---|
| Maximum level of respiratory support | None | 48 | 19.4 |
| | Intranasal oxygen | 166 | 67.2 |
| | Facemask oxygen | 19 | 7.7 |
| | Noninvasive ventilation | 8 | 3.2 |
| | Mechanical ventilation | 5 | 2.4 |
| Total | | 247 | 100 |
| Treatment received | Steroids | 226 | 91.5 |
| | Antibiotics | 245 | 99.2 |
| | Prophylactic anticoagulation | 68 | 27.5 |
| | Therapeutic anticoagulation | 169 | 68.4 |
| | Antiviral (Remdesivir) | 18 | 7.3 |

$CI_{95\%}$ = 1.04 to 3.75] and length of stay >10 days in the hospital [aOR = 2.62; $CI_{95\%}$ = 1.2 to 5.72] were found to be significantly associated with the outcome on multivariate analysis.

## Discussion

In this study, the prevalence of persistent symptoms in patients hospitalized for acute COVID-19 was assessed at ≥4 weeks after symptom onset. Among 247 patients included, 178 (72.1%) had one or more symptoms of Long COVID at a median follow-up time of 35 days (IQR 32–40). This number is within the range of results from several studies, with prevalence estimates varying from 6.2 to 80% [3, 6, 8–14]. While this wide range is due to multiple factors, one of the most important is the varying definition of Long COVID used in the studies. We get a slightly narrower estimate of 20 to 80% when we look at those studies that used a similar time cut-off to define Long COVID to this study [3, 7, 8].

In a prospective cohort study in South Africa by M. Dryden et al that evaluated patients at 1 month and 3 months after hospitalization, 82.1% had one or more persistent symptoms at 1 month [7]. Similarly, 80% of the patients had one or more long-term symptoms at follow-up

**Table 4. Symptoms reported during follow up visits.**

| Symptom | Frequency (n) | Percentage (%) |
|---|---|---|
| Fatigue | 103 | 41.7 |
| Shortness of breath | 77 | 31.2 |
| Cough | 67 | 27.1 |
| Sleep disturbances | 37 | 15 |
| Chest pain or pressure | 16 | 6.5 |
| $SpO_2$ <90 on room air | 9 | 3.6 |
| Headache | 8 | 3.2 |
| Decreased/absent sense of smell | 6 | 2.4 |
| Worsening symptoms after exertion | 4 | 4.6 |
| Palpitation | 3 | 1.2 |
| Dizziness | 3 | 1.2 |
| Respiratory crackles/wheeze | 2 | 0.8 |
| Non-specific pain | 2 | 0.8 |
| Anxiety | 1 | 0.4 |
| others | 76 | 30.8 |

**Table 5. Factors associated with Long COVID.**

| | | COR with 95% CI | p-value | AOR with 95% CI | p-value |
|---|---|---|---|---|---|
| **Age** | 18–69 | 1 | | | |
| | 70 or above | 2.241 (1.062, 4.728) | **0.034** | 2.064 (0.855, 4.984) | 0.107 |
| **Sex** | Male | 1 | | | |
| | Female | 1.123 (0.624, 2.019) | 0.699 | | |
| **Weight** | Normal | 1 | | | |
| | Overweight/obese | 1.701 (0.676, 4.282) | 0.259 | | |
| **Duration of viral shedding** | <2 weeks | 1 | | | |
| | >2 weeks | 1.859 (1.002, 3.448) | **0.049** | 1.366 (0.673, 2.773) | 0.387 |
| **Vaccination** | No | 1 | | | |
| | Yes | 0.567 (0.155, 2.074) | 0.391 | | |
| Severity* | Mild/moderate | 1 | | | |
| | Severe | 1.310 (0.593, 2.893) | 0.505 | | |
| | Critical | 4.452 (0.867, 22.878) | 0.074 | | |
| **Number of symptoms** | <5 | 1 | | | |
| | 5 or more | 1.572 (0.898, 2.749) | 0.113 | 1.433 (0.756, 2.718) | 0.270 |
| **Duration of symptoms before admission** | <7 | 1 | | | |
| | >7 | 2.051 (1.153, 3.65) | **0.015** | 1.971 (1.037, 3.748) | **0.038** |
| **Comorbidities** | Not present | 1 | | | |
| | 1 comorbidity | 0.671 (0.375, 1.201) | 0.179 | 0.570 (0.294, 1.106) | 0.097 |
| | 3 or more comorbidities | 1.423 (0.614, 3.298) | 0.410 | | |
| | Chronic respiratory illness | 0.759 (0.273, 2.109) | 0.597 | | |
| | Diabetes | 1.043 (0.579, 1.878) | 0.889 | | |
| | Hypertension | 1.342 (0.734, 2.455) | 0.340 | | |
| | CKD | 1.172 (0.308, 4.462) | 0.816 | | |
| | Chronic cardiac illness | 0.692 (0.245, 1.949) | 0.485 | | |
| **Duration of admission** | ≤10 days | 1 | | | |
| | >10 days | 2.908 (1.608, 5.259) | **<0.001** | 2.621 (1.200, 5.727) | **0.016** |
| | Odds for each additional day in the hospital | 1.083 (1.013, 1.158) | **0.019** | 0.986 (0.920, 1.056) | 0.681 |
| **ICU admission** | No | 1 | | | |
| | Yes | 4.241 (0.964, 18.654) | 0.056 | 1.840 (0.182, 18.631) | 0.605 |
| **Level of respiratory support** | None | 1 | | | |
| | Intranasal oxygen, facemask | 1.132 (0.569, 2.254) | 0.724 | | |
| | NIV/MV | 5.909 (0.707, 49.404) | 0.101 | 2.138 (0.091, 50.528) | 0.638 |
| **Complications during treatment** | No | 1 | | | |
| | Yes (Any complication) | 1.311 (0.750, 2.291) | 0.341 | | |
| | 2 or more complications | 2.629 (0.752, 9.185) | 0.13 | 1.821 (0.286, 11.587) | 0.526 |
| | Odds for each additional complication | 1.408 (0.927, 2.140) | 0.109 | 0.869 (0.448, 1.687) | 0.679 |

*Severity omitted from multivariate analysis because of high degree of correlation with ICU admission. Abbreviations: AOR–Adjusted odds ratio, CKD–Chronic Kidney Disease, COR–Crude odds ratio, MV–mechanical ventilation, NIV–noninvasive ventilation

days ranging from 14 to 110 days in a meta-analysis of 47,910 patients [8]. These results are similar to the findings in our study. On the other hand, other studies have reported much lower estimates. For instance, a meta-analysis on the global prevalence of Long COVID found a prevalence of 37% at 30 days. However, this study included participants who were not hospitalized during their acute infection [12].

Our study does not directly indicate the risk that any patient may experience persistent symptoms after contracting COVID-19, because the prevalence estimate only includes patients who were hospitalized and had follow-up visits at the time point mentioned above. Patients who recovered fully and did not need a follow-up visit at ≥4 weeks, or who did not have symptoms severe enough to necessitate hospitalization, are not included in this study and its conclusions do not apply to them.

In agreement with most other studies, we found that the most prevalent symptom of Long COVID is fatigue, followed by shortness of breath and cough [3, 6, 8–14]. Similarly, in a study from Ethiopia on post-COVID-19 pulmonary complications done by Abebaw B. et al, 17.8% of the participants had respiratory complaints at 3 months from hospital discharge [18]. The most common of these included cough (7.7%) and shortness of breath (14.1%). The differences in the prevalence with this study could be due to the later date of follow-up used. In addition, 5.7% of the patients had $SpO_2$ <90%, which was comparable to the finding in our study of 3.6% (Table 4).

Participants in this study had some persistent laboratory abnormalities including lymphopenia (17.8%) and raised C-reactive protein (CRP) (8.5%). In the above-mentioned study from Ethiopia, 7.9% of the participants had lymphopenia at 3 months. This lower estimate could be due to a longer follow-up time in the study. Similar to our finding, raised CRP was found in 8% of patients included in the abovementioned meta-analysis [8]. Other laboratory abnormalities that are commonly reported in similar studies, such as elevated D-dimer, ferritin, and NT-proBNP levels could not be assessed in this study (They were not monitored in the study participants because they are not readily available).

Another feature that was present in our study participants is persistent radiological imaging abnormalities. Fibrosis and ground-glass opacity were the most frequent findings on chest x-ray, being reported in 42.6% and 39.7% of the patients for whom chest x-ray was done respectively. This is consistent with data from other research worldwide [8–10]. This percentage may vary if all patients had gotten follow-up imaging, instead of only those with clinical indications.

Two factors were found to be significantly associated with Long COVID; these are duration of symptoms more than 7 days before admission and length of hospital stay more than 10 days. While the relationship between delayed diagnosis of COVID-19 and the severity of acute illness has been explored in various studies, none were found that looked for an association with Long COVID. The presence of a longer duration of symptoms before admission could reflect a delay in diagnosis, leading to delayed administration of treatment. This could possibly lead to a slower recovery.

Patients who stayed in the hospital for more than 10 days were more likely to have Long COVID than those who had a shorter stay. This implies patients who took longer to recover while they were admitted were also more likely to have continuing symptoms at 4 weeks after disease onset. If the extended hospital stay is a risk factor for Long COVID itself, or if the same reasons that led to a longer course in the hospital also led to Long COVID, it should be investigated further.

Neither age nor sex was significantly associated with having Long COVID in this study. Previous studies have yielded conflicting results regarding these factors, some reporting a significant association with older age [9, 13, 18], as well as female sex [6, 8–10, 13, 14], and others reporting none [3]. In this study, while those patients aged ≥70 were found to have 2.2 times increased odds of Long COVID on bivariate analysis, this was not true when controlling for confounders on multivariate analysis. This could suggest age alone may not confer risk for Long COVID.

Above-normal BMI (overweight/obesity) was not found to be significantly associated with Long COVID in this study. This was also true in a prospective cohort study from South Africa, which did not find obesity to be an important predictor of Long COVID [7]. On the other hand, other studies from Switzerland and the USA found obese patients to have increased odds of Long COVID [13]. Since obesity increases the risk of severe illness in the acute phase of COVID-19 (according to various meta-analyses and cohort studies [22]), a potential association with persistent symptoms in obese patients should be further investigated before conclusions can be made.

Other potential risk factors that were not found to have a significant association with Long COVID in this study include ICU admission, having one or more comorbidities (such as diabetes mellitus, chronic lung conditions), need for oxygen or respiratory support during the acute phase of the illness, and increased number of symptoms on initial presentation. In other studies, one or more of these factors were reported to have an impact on Long COVID, however, the relevance of these factors was not consistent across the investigations [3, 9, 10, 13, 14].

## Conclusion and recommendations

There was a high prevalence of Long COVID among adult patients who were followed after hospital discharge and at least 4 weeks following symptom onset. The most reported symptoms were fatigue, shortness of breath, and cough. Duration of symptoms before hospitalization and length of stay in the hospital were found to have a significant association with the outcome. Prospective studies are needed that assess patients' persistent symptoms for a longer duration of follow-up, and with standard tools to measure the severity of symptoms.

## Limitations

There are several limitations to this study. Since it was done in a single center, generalizing the results to the general population is challenging. In addition, owing to the retrospective design of the study, information that was not recorded in the charts may be missed. For example, information on the severity and duration of each symptom would have been valuable. Moreover, issues that may not be addressed during a routine clinic visit, such as quality of life, could not be assessed. In addition, due to the lack of a control group, we can't conclude that all the symptoms the patients are experiencing are due to Long COVID alone, and not a coexisting medical condition. Another drawback is that it is not possible to fully attribute all of the patients' findings to Long COVID due to the lack of data on the prevalence of each symptom prior to the COVID-19 infection.

## Supporting information

**S1 File. Supplementary tables.**
(DOCX)

**S2 File. Data collection checklist.**
(DOCX)

## Acknowledgments

We would like to acknowledge the frontline health workers who risked their lives to provide service during the COVID-19 pandemic.

## Author Contributions

**Conceptualization:** Bethlehem Berhanu Minassie, Wondwossen Amogne Degu, Dawit Kebede Huluka.

**Data curation:** Bethlehem Berhanu Minassie, Wondwossen Amogne Degu, Eyob Kebede Etissa, Natnael Fitsum Asfeha, Salem Taye Alemayehu, Dawit Kebede Huluka.

**Formal analysis:** Bethlehem Berhanu Minassie, Eyob Kebede Etissa.

**Funding acquisition:** Bethlehem Berhanu Minassie, Wondwossen Amogne Degu, Dawit Kebede Huluka.

**Investigation:** Bethlehem Berhanu Minassie, Natnael Fitsum Asfeha, Salem Taye Alemayehu, Dawit Kebede Huluka.

**Methodology:** Bethlehem Berhanu Minassie, Wondwossen Amogne Degu, Dawit Kebede Huluka.

**Project administration:** Bethlehem Berhanu Minassie, Natnael Fitsum Asfeha.

**Resources:** Bethlehem Berhanu Minassie, Natnael Fitsum Asfeha, Dawit Kebede Huluka.

**Software:** Bethlehem Berhanu Minassie, Eyob Kebede Etissa.

**Supervision:** Bethlehem Berhanu Minassie, Wondwossen Amogne Degu, Dawit Kebede Huluka.

**Validation:** Bethlehem Berhanu Minassie, Dawit Kebede Huluka.

**Visualization:** Bethlehem Berhanu Minassie, Dawit Kebede Huluka.

**Writing – original draft:** Bethlehem Berhanu Minassie.

**Writing – review & editing:** Bethlehem Berhanu Minassie, Wondwossen Amogne Degu, Eyob Kebede Etissa, Salem Taye Alemayehu, Dawit Kebede Huluka.

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
