## [Decision Letter · Decision Letter 0]

2 Jul 2024

PONE-D-24-21628Attributes and factors associated with Long COVID in patients hospitalized for acute COVID-19: A retrospective cohort studyPLOS ONE

Dear Dr. Minassie, 

Thank you for submitting your manuscript to PLOS ONE. After careful consideration, we feel that it has merit but does not fully meet PLOS ONE’s publication criteria as it currently stands. Therefore, we invite you to submit a revised version of the manuscript that addresses the points raised during the review process.

Kindly attend to all the comments made by the reviewers

We look forward to receiving your revised manuscript.

Kind regards,

Zivanai Cuthbert Chapanduka, MBChB (M.D)

Academic Editor

PLOS ONE

Journal Requirements:

3. We note you have included a table to which you do not refer in the text of your manuscript. Please ensure that you refer to Table 2 and 5 in your text; if accepted, production will need this reference to link the reader to the Table.

Additional Editor Comments:

Dear Dr Bethlehem Berhanu Minassie

Congratulations on making it to the second round of review. The reviewers require you to make minor/major revisions to your manuscript. Please reviewer , understand and act on all their requirements. In cases where you believe the reviewer misunderstood your manuscript, please explain this to the best of your ability. Please note that Plos follows a policy which allows maximum number of reviews before the manuscript has to be rejected for resubmission.

Thank you.

Kind regards

Prof Zivanai Cuthbert Chapanduka

Academic Editor

Reviewers' comments:

Reviewer's Responses to Questions

**Comments to the Author**

1. Is the manuscript technically sound, and do the data support the conclusions?

Reviewer #1: Yes

Reviewer #2: Partly

2. Has the statistical analysis been performed appropriately and rigorously? 

Reviewer #1: Yes

Reviewer #2: I Don't Know

3. Have the authors made all data underlying the findings in their manuscript fully available?

Reviewer #1: No

Reviewer #2: Yes

4. Is the manuscript presented in an intelligible fashion and written in standard English?

Reviewer #1: Yes

Reviewer #2: Yes

5. Review Comments to the Author

Reviewer #1: Dear Authors

Thank you for submitting your research for publication. This is a well-structured manuscript. However, there are areas for improvement. Please address the following points:

General comments

- The authors stated that patients who recovered fully and did not need a follow-up visit at ≥4 weeks were not included in this study. This, if unjustified, could significantly impact the study design as it may result in an overestimation of the prevalence of long covid among hospitalized patients in this cohort. Please ensure the following is clarified: - inclusion and exclusion criteria. - clarify if the excluded group of patients were fully recovered with no long covid symptoms or there is no data regarding long covid symptoms in this group.

- Before using an abbreviation for the first time make sure you mention the full term e.g. lung diffusion test for carbon monoxide (DLCO).

- Include all tables referenced in the manuscript (Table 6 is missing) and ensure the tables and figures are labelled properly/correctly.

- Address typo grammatical errors - There are typo-grammatical errors throughout the manuscript. I highlighted some of them on the pdf file.

- Each statement should be referenced.

Method:

Line 141-144: Please expand on inclusion and exclusion criteria for the study population. Justify reasons for exclusions.

Results:

Line 203: Several studies found a correlation between multiple comorbidities and long COVID. In this study, what is the correlation between long COVID and more than one preexisting comorbidities?

Line 226-227: " The most common abnormality was fibrosis, detected in 29 (42.6%) of the patients, followed by ground-glass opacities in 27 (39.7%) patients”. Consider rephrasing to clarify that the percentage is calculated from the patients who had imaging, not the total cohort.

Line 243: Table 6 is not included in the submission. Please add Table 6 to the manuscript.

Discussion:

Line 266-267: The authors stated that patients who recovered fully and did not need a follow-up visit at ≥4 weeks were not included in this study. As mentioned above, please clarify if these patients were fully recovered with no long covid symptoms or there is no data regarding long covid in this group.

Line 342: The limitations should be expanded to include a more detailed discussion of potential confounders.

Best wishes

Kind regards

Reviewer #2: The authors draw rational and reasonable conclusions from the results they have found using the methods described.

Major revisions are needed in the methods section to clarify some of the methodology used for this study. Also see other comments below.

Line 79: Define CDC and WHO.

Line 94: Define DLCO.

Line 95 and 273: Please specify sleep issues. ?insomnia.

Line 98: Abnormal imaging in which anatomical region? What type of imaging?

Line 102: Define CT.

(Please define all abbreviations used for the first time in the manuscript.)

Line 148: Please provide the pre-tested checklist that was developed as a supplementary document.

Line 155: Please provide city and country of software.

Line 156: Add ‘(IQR)’ after interquartile range which is later used as an abbreviation.

Line 138: Please specify the exclusion criteria. Perhaps it is better to illustrate the sampling procedure using a flowchart. How many patients were initially identified between March 2020 and December 2022? How many of these patients were then included and excluded based on which exclusion and inclusion criteria?

This also helps to clarify what follows. Line 143: “…having at least one follow-up four weeks after the onset of symptoms – were included.” But in the results sections it seems that patients could be included in the study even if they were not followed up at least four weeks after the onset of symptoms (line 210).

Table 2

• What scores and/or parameters were used to determine the severity of acute COVID-19 infection?

• What were the laboratory cut-offs used for each abnormal laboratory parameter? Also provide the unit of measurement for each. And were the same laboratory cut-offs used in the Ethiopian study mentioned in line 283?

• What were regarded as ‘atypical’ imaging abnormalities?

Line 227: Other abnormalities 0.1%? What is the n value?

Line 224: Did the patients with lymphopenia also have neutrophilia or neutropenia? And in the patients with leucocytosis, which of the white blood cells in the differential blood count were responsible for the leucocytosis?

Line 234: Authors refer to Table 6?

Line 237: Please define aOR and cOR.

Line 278: I am not sure where the authors mentioned the SpO2 results in their results section discussed here. If this is only shown in a table, please refer to the table in the text. How many of the patients who had SpO2 <90% had known chronic respiratory or cardiac disease?

Could the authors please indicate why the levels of important “commonly reported” markers such as D-dimer and NT-proBNP which has been linked with severity of disease “…could not be assessed in this study”?

Line 289: ^radiological imaging abnormalities.

Line 292: Why do the authors assume that a higher percentage would have abnormalities than a lower percentage if all patients had undergone CXR at follow-up?

Lines 300, 307: Please add the P values as well for ease of reference.

Line 304: Could the authors please specify what is meant by “downhill effects on recovery” and rephrase this in the manuscript for clarity.

6. PLOS authors have the option to publish the peer review history of their article (what does this mean?). If published, this will include your full peer review and any attached files.

Reviewer #1: No

Reviewer #2: No

---

## [Author Response · Author response to Decision Letter 0]

19 Aug 2024

Thank you for taking the time to assess our work. We appreciate the suggested modifications and have revised the manuscript accordingly. Here are our responses to each of your comments. 

Reviewer #1

General comments

The authors stated that patients who recovered fully and did not need a follow-up visit at ≥4 weeks were not included in this study. This, if unjustified, could significantly impact the study design as it may result in an overestimation of the prevalence of long covid among hospitalized patients in this cohort. Please ensure the following is clarified: - inclusion and exclusion criteria. - clarify if the excluded group of patients were fully recovered with no long covid symptoms or there is no data regarding long covid symptoms in this group.

Response:

We apologize for failing to make the methodology clear. The ‘sampling procedure and eligibility criteria’ section in the methodology has been modified and a flow chart is added to make the patient selection clearer. There were a total of 880 admitted patients. Some patients (n=132) were admitted despite being asymptomatic from the outset, and were only admitted for isolation. These patients were excluded. 

For those who had symptoms and received treatment, the hospital’s protocol was to have a follow-up visit once they are discharged. However, owing to the burden on healthcare facilities imposed by the pandemic, some patients whom the discharge physician assessed to be asymptomatic at discharge received a phone call follow-up rather than an in person visit. These patients were fully recovered upon discharge, however, they were not included the study. This is because only those patients of whom we have documented data at ≥4 weeks from symptom onset are able to be assessed for Long COVID (to meet the definition of Long COVID).

For your ease of reference, we have inserted the modified section of the methodology here.

To increase the power of the study, all patients who met the eligibility criteria—namely, age above 18, having a confirmed symptomatic SARS-CoV-2 infection, and having at least one follow-up visit four weeks after the onset of symptoms—were included. Patients whose follow-up ended before four weeks have elapsed from symptom onset were excluded from the study. 

Between March 2020 and December 2022, a total of 880 patients were admitted to Hallelujah General Hospital for acute COVID-19 treatment. Out of these, 739 patients were discharged from the hospital after completing their treatment/isolation period, and 247 of the discharged patients had in-person follow-up visit after four weeks. Among the remaining patients, 132 never developed any symptoms of COVID-19 and were not included in the study. And an additional 284 patients had post-discharge follow-up phone call and were reported to be asymptomatic, and hence were not appointed for further in-person visits (Fig 1).

Before using an abbreviation for the first time make sure you mention the full term e.g. lung diffusion test for carbon monoxide (DLCO).

Response:

This has been corrected. We apologize for this repeated mistake throughout the manuscript.

Include all tables referenced in the manuscript (Table 6 is missing) and ensure the tables and figures are labelled properly/correctly.

Response:

The labeling of the tables has been corrected. What was written as Table 6 was meant to be Table 5.

Address typo grammatical errors - There are typo-grammatical errors throughout the manuscript. I highlighted some of them on the pdf file.

Response:

These have been corrected.

Each statement should be referenced.

Response:

We found a statement in the Discussion section that was unreferenced and that has been corrected.

Method:

Line 141-144: Please expand on inclusion and exclusion criteria for the study population. Justify reasons for exclusions.

Response:

As discussed above, this section has been modified to show which patients have been excluded.

Line 203: Several studies found a correlation between multiple comorbidities and long COVID. In this study, what is the correlation between long COVID and more than one preexisting comorbidities?

Response:

In this study, no correlation was found between multiple comorbidities and Long COVID. Regression analyses were done to assess effects of having ≥3 and ≥4 comorbidities; however, no significant association was found. A line reflecting that result has been added in Table 5. 

Line 226-227: " The most common abnormality was fibrosis, detected in 29 (42.6%) of the patients, followed by ground-glass opacities in 27 (39.7%) patients”. Consider rephrasing to clarify that the percentage is calculated from the patients who had imaging, not the total cohort.

Response:

This statement has been rephrased to “Among patients for whom chest X-ray was obtained, the most common abnormality was fibrosis, detected in 29 (42.6%) patients, followed by ground-glass opacities in 27 (39.7%) patients. Other abnormalities included reticulations, consolidation, and pleural effusion.”

Line 243: Table 6 is not included in the submission. Please add Table 6 to the manuscript.

Response:

What was written as Table 6 was meant to be Table 5. This has been corrected.

Discussion:

Line 266-267: The authors stated that patients who recovered fully and did not need a follow-up visit at ≥4 weeks were not included in this study. As mentioned above, please clarify if these patients were fully recovered with no long covid symptoms or there is no data regarding long covid in this group

Response:

As discussed above, the patients who didn’t have follow up at 4 weeks were assessed to be fully recovered at discharge and received only phone call follow-up.

Line 342: The limitations should be expanded to include a more detailed discussion of potential confounders.

Response:

The limitations have been expanded.

Reviewer #2

Line 79: Define CDC and WHO.

Response:

This has been done. Other abbreviations have also been defined when used for the first time in the manuscript. We apologize for this repeated mistake throughout the manuscript.

Line 94: Define DLCO.

Response:

Done

Line 95 and 273: Please specify sleep issues. ?insomnia.

Response:

The term ‘sleep issues’ has been rephrased to ‘disturbed sleep’. This is meant to encompass mainly insomnia but also fragmentation of sleep and non-restorative sleep. Upon our literature review, similar studies and meta-analyses report disturbed sleep as one of the main symptoms of Long COVID, which is why we chose in include it in our assessment. 

Line 98: Abnormal imaging in which anatomical region? What type of imaging? 

Response:

“Abnormal imaging…”has been rephrased to “Abnormal chest X-ray/computed tomography [CT]…”

Line 102: Define CT.

(Please define all abbreviations used for the first time in the manuscript.)

Response:

Corrected

Line 148: Please provide the pre-tested checklist that was developed as a supplementary document

Response:

This has been done as a file named S2.

Line 155: Please provide city and country of software.

Response:

This has been added as SPSS (New york, USA). If we have misunderstood your comment, please let us know and we will correct it again. 

Line 156: Add ‘(IQR)’ after interquartile range which is later used as an abbreviation.

Response:

Corrected

Line 138: Please specify the exclusion criteria. Perhaps it is better to illustrate the sampling procedure using a flowchart. How many patients were initially identified between March 2020 and December 2022? How many of these patients were then included and excluded based on which exclusion and inclusion criteria?

This also helps to clarify what follows. Line 143: “…having at least one follow-up four weeks after the onset of symptoms – were included.” But in the results sections it seems that patients could be included in the study even if they were not followed up at least four weeks after the onset of symptoms (line 210).

We apologize for failing to make the methodology clear. The ‘sampling procedure and eligibility criteria’ section in the methodology has been modified and a flow chart is added to make the patient selection clearer. There were a total of 880 admitted patients. Some patients (n=132) were admitted despite being asymptomatic from the outset, and were only admitted for isolation. These patients were excluded. 

For those who had symptoms and received treatment, the hospital’s protocol was to have a follow-up visit once they are discharged. However, owing to the burden on healthcare facilities imposed by the pandemic, some patients whom the discharge physician assessed to be asymptomatic at discharge received a phone call follow-up rather than an in person visit. These patients were fully recovered upon discharge, however, they were not included the study. This is because only those patients of whom we have documented data at ≥4 weeks from symptom onset are able to be assessed for Long COVID (to meet the definition of Long COVID).

For your ease of reference, we have inserted the modified section of the methodology here.

To increase the power of the study, all patients who met the eligibility criteria—namely, age above 18, having a confirmed symptomatic SARS-CoV-2 infection, and having at least one follow-up visit four weeks after the onset of symptoms—were included. Patients whose follow-up ended before four weeks have elapsed from symptom onset were excluded from the study. 

Between March 2020 and December 2022, a total of 880 patients were admitted to Hallelujah General Hospital for acute COVID-19 treatment. Out of these, 739 patients were discharged from the hospital after completing their treatment/isolation period, and 247 of the discharged patients had in-person follow-up visit after four weeks. Among the remaining patients, 132 never developed any symptoms of COVID-19 and were not included in the study. And an additional 284 patients had post-discharge follow-up phone call and were reported to be asymptomatic, and hence were not appointed for further in-person visits (Fig 1).

Table 2

• What scores and/or parameters were used to determine the severity of acute COVID-19 infection?

Severity of acute COVID-19 infection was defined as per the national COVID-19 management guideline. A table showing the classification criteria of acute COVID-19 into mild, moderate, severe, and critical has been inserted in the supplementary S1 file. The reference is added as well. 

• What were the laboratory cut-offs used for each abnormal laboratory parameter? Also provide the unit of measurement for each. And were the same laboratory cut-offs used in the Ethiopian study mentioned in line 283?

The lab cut-offs and units of measurement have been added. And yes, the same cut-offs were used in the Ethiopian study mentioned.

• What were regarded as ‘atypical’ imaging abnormalities?

Typical, atypical, and indeterminate imaging abnormalities were based on the Radiological Society of North America (RSNA) recommendations for reporting COVID-19 chest imaging findings. This has been stated in the operational definitions, and the descriptions are provided in the supplementary S1 file.

Line 227: Other abnormalities 0.1%? What is the n value?

Response:

We could not find the sentence you are referring to. Can you please clarify the comment?

Line 224: Did the patients with lymphopenia also have neutrophilia or neutropenia? And in the patients with leucocytosis, which of the white blood cells in the differential blood count were responsible for the leucocytosis?

Response:

The patients with lymphopenia had either neutrophilia or normal neutrophil counts. In the patients with leukocytosis, this was mainly driven by increased neutrophil counts.

Line 234: Authors refer to Table 6?

Response:

We apologize for this mistake. This was meant to read Table 5.

Line 237: Please define aOR and cOR.

Response:

Corrected

Line 278: I am not sure where the authors mentioned the SpO2 results in their results section discussed here. If this is only shown in a table, please refer to the table in the text. How many of the patients who had SpO2 <90% had known chronic respiratory or cardiac disease?

Response:

This result is given in table 4 and it is now referred to in the text. This statement is also added to address this comment. Nine patients (3.6%) had SpO2 less than 90% on room air, out of which 2 patients had a preexisting cardiopulmonary illness.

Could the authors please indicate why the levels of important “commonly reported” markers such as D-dimer and NT-proBNP which has been linked with severity of disease “…could not be assessed in this study”?

Response:

This is because these tests are not widely available and were not done for the patients. The statement has been revised as follows to make this clarification. Other laboratory abnormalities that are commonly reported in similar studies, such as elevated D-dimer, ferritin, and NT-proBNP levels could not be assessed in this study (They were not monitored in the study participants because they are not readily available).

Line 289: ^radiological imaging abnormalities.

Response:

Corrected

Line 292: Why do the authors assume that a higher percentage would have abnormalities than a lower percentage if all patients had undergone CXR at follow-up?

Response:

Thank you for pointing out this mistake. We would expect a lower percentage and not a higher one if all patients had been imaged. The statement has been modified as follows. This percentage may vary if all patients had gotten follow-up imaging, instead of only those with clinical indications.

Lines 300, 307: Please add the P values as well for ease of reference.

Response:

P values have been added as suggested.

Line 304: Could the authors please specify what is meant by “downhill effects on recovery” and rephrase this in the manuscript for clarity.

Response:

This statement has been modified to ‘This delay could possibly lead to a slower recovery.’ We mean that those patients who had symptoms for >7 days before being diagnosed would start treatment later than those who were diagnosed earlier, and this might lead to a slower recovery. 

Thank you for evaluating our manuscript. We have tried to address your concerns and believe that our paper has improved considerably. We would be happy to make further corrections if necessary and look forward to hearing from you soon.

---

## [Decision Letter · Decision Letter 1]

22 Sep 2024

PONE-D-24-21628R1Attributes and factors associated with Long COVID in patients hospitalized for acute COVID-19: A retrospective cohort studyPLOS ONE

Dear Dr. Minassie,

Thank you for submitting your manuscript to PLOS ONE. After careful consideration, we feel that it has merit but does not fully meet PLOS ONE’s publication criteria as it currently stands. Therefore, we invite you to submit a revised version of the manuscript that addresses the points raised during the review process.

We look forward to receiving your revised manuscript.

Kind regards,

Zivanai Cuthbert Chapanduka, MBChB (M.D)

Academic Editor

PLOS ONE

Journal Requirements:

Reviewers' comments:

Reviewer's Responses to Questions

**Comments to the Author**

1. If the authors have adequately addressed your comments raised in a previous round of review and you feel that this manuscript is now acceptable for publication, you may indicate that here to bypass the “Comments to the Author” section, enter your conflict of interest statement in the “Confidential to Editor” section, and submit your "Accept" recommendation.

Reviewer #1: All comments have been addressed

Reviewer #2: All comments have been addressed

2. Is the manuscript technically sound, and do the data support the conclusions?

Reviewer #1: Yes

Reviewer #2: Yes

3. Has the statistical analysis been performed appropriately and rigorously? 

Reviewer #1: Yes

Reviewer #2: Yes

4. Have the authors made all data underlying the findings in their manuscript fully available?

Reviewer #1: Yes

Reviewer #2: Yes

5. Is the manuscript presented in an intelligible fashion and written in standard English?

Reviewer #1: Yes

Reviewer #2: Yes

6. Review Comments to the Author

Reviewer #1: (No Response)

Reviewer #2: The authors have improved their manuscript.

Please see minor revisions required:

Methods

Line 145: Is this the calculated sample size required for statistical significance? Please clarify this with a clear statement.

Line 165-167: Reference the file named S2 here.

Results

From previous review Line 227: Other abnormalities 0.1%? What is the n value?

Response:

We could not find the sentence you are referring to. Can you please clarify the comment?

- How many of the patients in the cohort had these “other abnormalities” that the authors are referring to?

Discussion

For all paragraphs in the Discussion section: Authors should avoid repeating the results in the Discussion section and instead only formulate discussion points around the results that are already included in the relevant section.

Line 290-295: Are the authors still referring to the study mentioned in the previous paragraph [12] as “This study”?

7. PLOS authors have the option to publish the peer review history of their article (what does this mean?). If published, this will include your full peer review and any attached files.

Reviewer #1: No

Reviewer #2: **Yes: **Ethan James Gantana

---

## [Author Response · Author response to Decision Letter 1]

27 Oct 2024

We thank the reviewers for taking the time to assess our work for a second time. We appreciate the suggested modifications and have revised the manuscript accordingly. 

In our previous revision, we edited our reference list and failed to state our reason why. We apologize for not clarifying why we did so. During our revision, we noted that we have mistakenly put the wrong reference on number 21 on the list. The paper we originally put on number 21 has not been retracted. However, we simply had listed the wrong paper out of human error. We have now rechecked our reference to make sure it is correct.

We have addressed each comment below. 

Reviewer 

Line 145: Is this the calculated sample size required for statistical significance? Please clarify this with a clear statement.

Response: A statement to clarify that this minimum sample size was required for statistical significance is added.

Line 165-167: Reference the file named S2 here.

Response: Done

From previous review Line 227: Other abnormalities 0.1%? What is the n value?

How many of the patients in the cohort had these “other abnormalities” that the authors are referring to?

Response: The number of patients that had consolidation, reticulation and pleural effusion has been added as follows.

Additional abnormalities included pleural effusion in one patient, consolidation in two patients, and reticulations in six patients. 

For all paragraphs in the Discussion section: Authors should avoid repeating the results in the Discussion section and instead only formulate discussion points around the results that are already included in the relevant section.

Response: We have removed sentences that were just repetitions of the results and didn’t serve a purpose to the discussion.

Line 290-295: Are the authors still referring to the study mentioned in the previous paragraph [12] as “This study”?

Response: The wording has been changed to “our study” to indicate we are referring to our study and not the one mentioned above.

We would like to once again thank the reviewers for evaluating our manuscript. We would be happy to make further corrections if necessary and look forward to hearing from you soon.

---

## [Editor Report · Decision Letter 2]

12 Nov 2024

PONE-D-24-21628R2Attributes and factors associated with Long COVID in patients hospitalized for acute COVID-19: A retrospective cohort studyPLOS ONE

Dear Dr. Minassie,

Thank you for submitting your manuscript to PLOS ONE. After careful consideration, we feel that it has merit but does not fully meet PLOS ONE’s publication criteria as it currently stands. Therefore, we invite you to submit a revised version of the manuscript that addresses the points raised during the review process.

We look forward to receiving your revised manuscript.

Kind regards,

Zivanai Cuthbert Chapanduka, MBChB (M.D)

Academic Editor

PLOS ONE

Journal Requirements:

Additional Editor Comments:

Dear Dr Birhanu

Kindly attend to the minor revisions as requested by the peer reviewer. Please ensure that the reviewers concerns are fully addressed because this is the last review round. As per Plos 1 policy, if the author does not fully meet the reviewers' requirements, the manuscript will have to be resubmitted.

Thank you

Kind regards

---

## [Author Response · Author response to Decision Letter 2]

24 Dec 2024

We thank the reviewers for taking the time to assess our work again. We appreciate the suggested modifications and have revised the manuscript accordingly. The revised actions are shown in boldface type, and all revisions in the manuscript are marked using the Word “track changes” feature. Sections of the manuscript added to this letter are shown in italics.

In our previous revision, we edited our reference list and failed to state our reason why. We apologize for not clarifying why we did so. During our revision, we noted that we have mistakenly put the wrong reference on number 21 on the list. The paper we originally put on number 21 has not been retracted. However, we simply had listed the wrong paper out of human error. We have now rechecked our reference to make sure it is correct.

We have addressed each comment below. 

Reviewer 

Line 145: Is this the calculated sample size required for statistical significance? Please clarify this with a clear statement.

Response: A statement to clarify that this minimum sample size was required for statistical significance is added.

Line 165-167: Reference the file named S2 here.

Response: Done

From previous review Line 227: Other abnormalities 0.1%? What is the n value?

How many of the patients in the cohort had these “other abnormalities” that the authors are referring to?

Response: The number of patients that had consolidation, reticulation and pleural effusion has been added as follows.

Additional abnormalities included pleural effusion in one patient, consolidation in two patients, and reticulations in six patients. 

For all paragraphs in the Discussion section: Authors should avoid repeating the results in the Discussion section and instead only formulate discussion points around the results that are already included in the relevant section.

Response: We have removed sentences that were just repetitions of the results and didn’t serve a purpose to the discussion.

Line 290-295: Are the authors still referring to the study mentioned in the previous paragraph [12] as “This study”?

Response: The wording has been changed to “our study” to indicate we are referring to our study and not the one mentioned above.

We would like to once again thank the reviewers for evaluating our manuscript. We have been notified this is the last round of comments, and we hope we have addressed all your concerns satisfactorily.

---

## [Editor Report · Decision Letter 3]

31 Dec 2024

Attributes and factors associated with Long COVID in patients hospitalized for acute COVID-19: A retrospective cohort study

PONE-D-24-21628R3

Dear Dr. Minassie

We’re pleased to inform you that your manuscript has been judged scientifically suitable for publication and will be formally accepted for publication once it meets all outstanding technical requirements.

Kind regards,

Zivanai Cuthbert Chapanduka, MBChB (M.D)

Academic Editor

PLOS ONE
---

## [Editor Report · Acceptance letter]

6 Jan 2025

PONE-D-24-21628R3 

PLOS ONE

Dear Dr. Minassie, 

I'm pleased to inform you that your manuscript has been deemed suitable for publication in PLOS ONE. Congratulations! Your manuscript is now being handed over to our production team.

Kind regards, 

on behalf of

Professor Zivanai Cuthbert Chapanduka 

Academic Editor

PLOS ONE